

# The acidic latex protein from *Hevea brasiliensis* serves as an anionic antimicrobial peptide

Methaporn Meethong[1], Kitiya Ekchaweng[1,2], Sumalee Obchoei[1,2], Chanawee Jakkawanpitak[1,2] and Phanthipha Runsaeng[1,2]

[1] Division of Health and Applied Sciences, Faculty of Science, Prince of Songkla University, Hatyai, Songkhla, Thailand
[2] Center of Excellence for Biochemistry, Faculty of Science, Prince of Songkla University, Hatyai, Songkhla, Thailand

## ABSTRACT

**Background**. Hev b5 is a unique acidic protein identified as an allergen in natural latex and latex gloves, known for stimulating histamine release from human basophils sensitized with serum from latex-allergic individuals. It is rich in glutamic acid and proline residues arranged in repeated motifs. The protein's unusual amino acid composition includes 48% negatively charged residues and 13% positively charged residues.

**Methods**. The recombinant form of Hev b5 (rHev b5) was produced in *Escherichia coli*. Its chitinase activity, which may provide antifungal properties by breaking down chitin in phytopathogen cell walls, was assessed. Additionally, the antibacterial activity of rHev b5 against Gram-positive and Gram-negative bacteria, including *Bacillus cereus*, *Staphylococcus aureus*, *E. coli* and *Salmonella typhi*, was evaluated. The potential enhancement of this activity in the presence of calcium or zinc ions was investigated to understand the underlying mechanism involving binding to microbial membranes via metal ion-mediated cationic salt bridges.

**Results**. rHev b5 exhibited significant chitinase activity and demonstrated substantial antibacterial effects against both Gram-positive and Gram-negative bacteria. The antibacterial activity was notably enhanced in the presence of zinc or calcium ions, suggesting that rHev b5 binds to microbial membranes through metal ion-mediated cationic salt bridges, leading to cell lysis and microbial death.

**Conclusion**. Antimicrobial properties and chitinase activity of Hev b5 underline its potential as an anionic antimicrobial peptide, offering both antifungal and antibacterial defenses. These findings position Hev b5 as a promising candidate for further research in antimicrobial peptide applications.

## INTRODUCTION

Natural rubber latex, derived from the rubber tree, *Hevea brasiliensis*, is a crucial material extensively utilized across various industries, including medical, automotive, and consumer products (*Nair, 2020*; *Guerra et al., 2021*). Proteins in natural rubber latex are embedded in

Corresponding author
Phanthipha Runsaeng,
phanthipha.r@psu.ac.th

the rubber particles and dissolved in the serum phase. These proteins, which can number in the hundreds, are integral to the structural integrity and mechanical properties of the latex (*Wagner & Breiteneder, 2005*; *Sussman, Beezhold & Kurup, 2002*). Specific proteins in latex can trigger Type I hypersensitivity reactions in susceptible individuals, leading to symptoms that range from mild skin irritations to severe anaphylactic reactions (*Buss, Kupek & Fröde, 2008*; *Pecquet, Leynadier & Dry, 1990*; *Hamilton, 2010*). The prevalence of latex allergies, particularly noted among healthcare workers and people with frequent latex exposure, has led to significant health concerns and regulatory measures aimed at reducing protein levels in latex products (*Koh et al., 2005*; *Slater, 1994*; *Slater, 1997*; *Turjanmaa, Reunala & Räsänen, 1988*; *Turjanmaa et al., 2000*; *Hnizdo et al., 2001*).

Natural rubber latex contains over 200 proteins, but only a subset of these are known to be allergenic. Some of the most notable allergenic proteins include Hev b1, Hev b3, Hev b5, Hev b 6.02, among others, classified according to their sequence homology and biological function (*Palosuo, 1997*; *Meade, Weissman & Beezhold, 2002*; *Cremer, Rihs & Raulf-Heimsoth, 2008*; *Sussman, Beezhold & Kurup, 2002*). Hev b5 (acidic latex protein) is one of the proteins rich in glutamic acid and is the most acidic in the cytoplasm of laticifer cells in the rubber tree, indicating that this allergen has the potential to trigger allergic reactions in humans. Mass spectrometry shows that Hevb5 has a molecular weight of 16 kDa and an isoelectric point (pI) of 3.5 (*Slater et al., 1996*). The protein can be found in ammonia-mixed solutions, heat-processed latex, and extracts from gloves. Even though it has been sterilized in an autoclave, this protein can still act as an allergen (*Yeang et al., 2002*). Moreover, it has been found that the Hevb5 protein triggers the release of histamine from basophils in patients allergic to latex, depending on the amount of exposure. The amino acid sequence of Hev b5 inferred from natural rubber latex shows high similarity to an acidic protein identified in kiwi fruit (*Actinidia deliciosa* var. *deliciosa*) (*Akasawa et al., 1996*).

In the study by *Martin (1991)*, it was found that the amino acid sequence of the acidic protein in natural rubber latex is similar to acidic chitinase from cucumber (*Cucumis sativus*) and chitinase/lysozymes from Virginia creeper (*Parthenocissus quinquifolia*). It was also found that this protein exhibits chitinase activity, an important feature of chitinase enzymes, which is the ability to break down chitin. Chitin is a carbohydrate that forms part of the cell walls of many phytopathogens, such as fungi (*Zarei et al., 2011*), demonstrating its antifungal activity. Additionally, a study by *Kanokwiroon et al. (2008)* investigated the antimicrobial effects of proteins purified from the rubber tree latex against oral pathogens using proteins extracted from the bottom fraction (B-serum) and centrifuged serum (C-serum) portions. It was found that proteins extracted from the B-serum portion could inhibit the growth of certain oral pathogens, especially *Candida* spp. It is possible that the allergenic protein Hev b5 from natural rubber latex is a protein with chitinase enzyme properties, which leads to antimicrobial activity. Therefore, in this study, we aim to investigate the properties of the aforementioned allergenic protein, especially its biological role related to inhibiting the growth of pathogens, including fungi, Gram-negative, and Gram-positive bacteria.

## MATERIALS & METHODS

### Bacterial strains and media

The bacterial strains utilized in this study, namely *Bacillus cereus*, *Escherichia coli*, *Staphylococcus aureus* and *Salmonella typhi*, were maintained as glycerol stocks at −80 °C. Prior to use, the bacterial strains were cultured on 1.5% (w/v) Luria-Bertani (LB) agar (Difco™ & BBL™, Sparks, USA) at 37 °C. *Phytophthora palmivora*, was isolated and cultured on 1.5% (w/v) potato dextrose agar (PDA) (Difco™ & BBL™, Sparks, USA) at 25 °C for the chitinase activity experiment.

The colloidal chitin for the chitinase detection medium was prepared following the method outlined by *Roberts & Selitrennikoff (1988)* as follows: Chitin flakes sourced from shrimp shells (Himedia™, Maharashtra, India) were hydrolyzed using 37% (w/w) cold HCl and stirred at 4 °C overnight. Subsequently, the mixture was filtered, and extraction was carried out in 95% cold ethanol at 4 °C overnight. The resulting chitin pellet, obtained through centrifugation, was washed with deionized water until the pH reached 7.0 and stored at 4 °C until use. The chitinase detection medium was prepared by combining 0.3 g $MgSO_4 \bullet 7H_2O$, 3.0 g $(NH_4)_2SO_4$, 2.0 g $KH_2PO_4$, 1.0 g $C_6H_8O_7 \bullet H_2O$, 15 g agar, 200 μL Tween-80, 4.5 g colloidal chitin, and 0.15 g bromocresol purple in one L of distilled water, adjusting the pH to 4.7, and autoclaving the mixture.

### Construction of the expression plasmid

The open reading frame (*orf*) of Hev b5, obtained from GenBank with accession number NM_001405317.1, was synthesized and ligated into the *Eco*RI and *Xho*I sites of the pET-28a(+) expression vector (GenScript, Jiangsu, China), which harbors the coding sequence for a hexa-histidine (6xHis) affinity tag. The resulting expression construct, named pET-28a(+) –Hev b5, was introduced into competent DH5α *E. coli* cells to enhance the plasmid copy number. Plasmids were then isolated from DH5α *E. coli* cells using the Plasmid Mini Prep Kit (ver. 2.0) (BioFACT, Daejeon, Korea), following the manufacturer's instructions. Validation of the plasmids was performed through nucleotide sequence analysis.

### Sequence analysis of Hev b5

The nucleotide sequence of an *orf* was translated into the amino acid sequences of Hev b5 and analyzed using Expasy tools. The potential signal peptide was identified by the SignalP 4.1 program. The protein domain characteristics were verified using the Simple Modular Architecture Research Tool (SMART) 7.0. *N*- and *O*-glycosylation sites were predicted using the NetNGlyc 1.0 and NetOGlyc 4.0. The tertiary structure of Hev b5 was modeled using the SWISS-MODEL program and visualized with PyMOL.

### Expression and extraction of recombinant Hev b5

For the expression of Hev b5, the plasmids containing pET-28a(+)—Hev b5 were introduced into competent BL21 Star™ (DE3) *E. coli* cells. A 1 ml volume of the overnight-grown culture was inoculated into 100 ml of LB broth supplemented with kanamycin. The culture was then incubated at 37 °C with agitation at 200 rpm until the optical density at

600 nm (O.D.$_{600}$) reached 0.45, at which point protein expression was induced by adding isopropyl β-D-thiogalactopyranoside (IPTG) (Vivantis, Selangor darul Ehsan, Malaysia) to a final concentration of 0.5 mM. After induction and overnight incubation, the cells were harvested by centrifugation at 5,000× g for 30 min at 4 °C.

The extraction of recombinant Hev b5 (rHev b5) was performed following the protocol described by *Peleg & Unger (2011)*, with minor modifications to suit the specific conditions of this study. Protein extraction was carried out as follows: bacterial cell walls were disrupted by employing 5 ml of lysis buffer (50 mM NaH$_2$PO$_4$, 300 mM NaCl, and 10 mM imidazole, pH 7.4) per milligram of wet weight, supplemented with 2.5 μg/ml of lysozyme. Additionally, sonication with a microtip was performed using a set amplitude of 25A, alternating between 30-second intervals of activation and deactivation, repeated 5 times. To enhance the quality of the extraction, RNase A and DNase I were added to final concentrations of 10 and 5 μg/ml, respectively, and incubated at 37 °C for 15 min. The cell lysate was then centrifuged at 10,000× g for 20 min at 4 °C. The supernatant was collected, while the pellet was resuspended in solubilization buffer (8 M urea, 0.1 M Tris-HCl, pH 8.0, 10 mM DTT). Subsequently, the soluble fraction and inclusion bodies were dialyzed against refolding buffer (0.1 M Tris-HCl, pH 8.0, 5 mM EDTA, 5 mM cysteine) with stirring at 4 °C overnight.

## Purification of rHev b5

The purification process of rHev b5 involved fast protein liquid chromatography (FPLC) (ÄKTA start, Uppsala, Sweden) utilizing a Ni-NTA HisTrap™ FF 1 ml column packed with Ni Sepharose™ 6 Fast Flow, which selectively binds to the 6xHis tag of rHev b5. Binding buffer (20 mM NaH$_2$PO$_4$, 0.5 M NaCl, pH 7.4) and rHev b5 were injected through the sample loop at a constant flow rate of 0.5 ml/min. Elution was achieved by applying a gradient of elution buffer (20 mM NaH$_2$PO$_4$, 0.5 M NaCl, 500 mM imidazole, pH 7.4), allowing the separation of fractions, each collected at 1 ml/tube. Subsequently, fractions containing rHev b5 were analyzed *via* 12% SDS-PAGE, and their concentrations were determined using the bicinchoninic acid (BCA) protein assay kit (Bio Basic, Markham, Ontario, Canada). The purified rHev b5 band was confirmed by Western blot analysis. Briefly, the protein was separated by 12% SDS-PAGE and electrophoretically transferred onto a polyvinylidene fluoride (PVDF) membrane. The membrane was blocked with 10% skim milk in TBS (25 mM Tris-HCl, pH 7.5, 0.5 M NaCl) and incubated with an anti-rHevb5 antibody (or anti-His antibody), produced by GenScript (1:2,500 dilution). After washing the membrane, peroxidase-conjugated goat anti-rabbit IgG (1:20,000 dilution) was added. The target protein band was visualized by a colorimetric reaction using 1-Step™ TMB-Blotting Substrate Solution (Thermo Fisher Scientific, Waltham, MA, USA), and the reaction was stopped by washing with water for 5 min.

## Chitinase activity assay

The chitinase activity of rHev b5 was initially assessed using the gel diffusion method on chitinase-selective media described by *Agrawal & Kotasthane (2012)*. The appearance of red-purple diffusion zones on the media, indicated by the color change of bromocresol

purple in response to pH variations, served as an indicator of chitinase activity. The chitinase-selective media were spread with 450 µg of rHev b5. Subsequently, *P. palmivora*, previously isolated and cultivated on 1.5% (w/v) PDA at 25 °C for 72 h, was transferred onto the selective media and further incubated at the same temperature for an additional 72 h. The extent of diffusion zones was documented with photographs, and the data were presented in terms of the annular radius value of the diffusion zone (mm), reflecting the red-purple coloration on the selective media. As controls, the experiment was repeated using 50 mM Tris-HCl, pH 8.0, and 1x Anti-Anti as negative controls, while 0.25 mg BSA was used as the internal control group.

The sample, standard, control, and blank tubes were prepared according to the chitinase activity assay protocol provided by Abbexa, ensuring that each tube contained a reaction volume of 80 µl. A volume of 80 µl of substrate was added to the sample and control tubes, and the contents were mixed thoroughly. The tubes were then incubated at 37 °C for 1 h and centrifuged at 5,000× g for 10 min at 4 °C. After centrifugation, 80 µl of the supernatant was transferred to new tubes and mixed with 40 µl of reaction buffer. The mixtures were heated in boiling water for 7 min, and the tubes were subsequently centrifuged at 5,000× g for 2 min. A volume of 80 µl of the resulting supernatant was transferred into each well of a microplate and mixed with 120 µl of dye reagent solution. The plate was incubated at 37 °C for 1 h. The absorbance was then measured and recorded at 585 nm. Chitinase activity was calculated, where one unit of chitinase activity is defined as the amount of enzyme required to produce one µg of *N*-acetylglucosamine per hour at 37 °C.

## Fungal growth inhibition assay

To evaluate the inhibitory potential of rHev b5 on fungal growth, *P. palmivora*, a common pathogenic fungus, was used in this study following a method adapted from *Sowanpreecha & Rerngsamran (2018)*. rHev b5 was serially diluted two-fold to final concentrations of 70, 35, 17.5, 8.75, and 0 µM in 50 mM Tris-HCl buffer, pH 8.0 with a total volume of one ml, before being combined with 4 ml of PDB. *P. palmivora*, pre-cultured on 1.5% (w/v) PDA at 25 °C for 120 h, was then introduced into each tube. Fungal growth was assessed by measuring the optical density at 600 nm following a 120-hour incubation at 25 °C. A parallel experiment was conducted using 50 mM Tris-HCl buffer, pH 8.0 without rHev b5, serving as the negative control.

To assess the inhibitory effect of rHev b5 on fungal growth, *P. palmivora*, a common pathogenic fungus, was used in this study. The procedure was adapted from *Hendricks, Christman & Roberts (2017)*. Briefly, rHev b5 was prepared at concentrations of 70 and 35 µM in 50 mM Tris-HCl (pH 8.0) and spread (100 µL) onto 1.5% (w/v) potato dextrose agar (PDA). *P. palmivora*, pre-cultured on 1.5% PDA at 25 °C for 7 days, was inoculated at the center of the plate using a No. 5 cork borer. The plates were then incubated at 25 °C for 2 and 5 days, and fungal growth inhibition was assessed by measuring radial inhibition (Ra) and capturing images. A parallel experiment was conducted using 50 mM Tris-HCl buffer (pH 8.0) without rHev b5 as the negative control. The percentage of growth inhibition was

calculated using the following formula:

$$\%I = [(Ra_c - Ra_t)/Ra_c] \times 100.$$

where $Ra_c$ is the mean colony diameter in the control group (assuming circular growth), and $Ra_t$ is the mean colony diameter in the test group (*Astiti & Suprapta, 2012*).

### Liquid growth inhibition assay

To assess the basic inhibitory capability of rHev b5 against bacteria (*Zheng et al., 2023*), both Gram-positive bacteria such as *B. cereus* and *S. aureus*, and Gram-negative bacteria such as *E. coli* and *S. typhi* were diluted 1:100 in LB broth during the mid-logarithmic phase for the experiment. The rHev b5 solution, at a concentration of 5.00 mg/ml, was serially diluted two-fold in a 96-well plate containing 50 mM Tris-HCl, pH 8.0, with a final volume of 100 µl in each well. Subsequently, 100 µl of the previously prepared bacterial suspension was added to each well. The plate underwent incubation at 37 °C with continuous agitation at 200 rpm, and bacterial growth was assessed every hour over a period of 16 h by measuring absorbance at 600 nanometers using the SPECTROstar Nano absorbance reader (BMG LABTECH, Ortenberg, Germany). LB broth served as the control group and underwent the same experimental conditions described above.

### Colony counting method

The colony counting method, adapted from *Scilletta et al. (2021)*, was employed to evaluate the antibacterial efficacy of rHev b5 at a specific time point. Mid-logarithmic phase cultures of both Gram-positive bacteria (*B. cereus*, *S. aureus*) and Gram-negative bacteria (*E. coli*, *S. typhi*) grown in LB broth were diluted at a ratio of 1:100 and then incubated with 70 nmol of rHev b5 at 37 °C for 12 h, with agitation at 200 rpm. Subsequently, the culture media were collected after specified durations and subjected to 10-fold serial dilution with 50 mM Tris-HCl, pH 8.0 until an appropriate dilution was achieved for subsequent measurements. Eventually, the bacterial suspensions were spread plated on 1.5% (w/v) LB agar and incubated at 37 °C for 16 h. The bacterial colonies within the range of 25 to 250 (*Tomasiewicz et al., 1980*) were counted and expressed as colony forming units per milliliter (CFU/ml). LB broth and ampicillin were employed as representatives of the negative and positive control groups, respectively.

### Bacterial binding activity of rHev b5

To verify that rHev b5 functions as an antimicrobial peptide through binding with microorganisms, the bacteria-binding assay described by *Huo et al. (2024)* was modified and conducted to suit the requirements of this study. *S. aureus* and *S. typhi* were selected to represent Gram-positive and Gram-negative bacteria, respectively, and were cultured. Then, 10 micrograms of rHev b5 was incubated with microorganisms ($1 \times 10^6$ cells) in TBS buffer containing 50 mM $CaCl_2$ or $ZnCl_2$ at room temperature for 20 min with rotation. The cells were pelleted by centrifugation at $10,000 \times$ g for 1 min, and the supernatant was collected as the S fraction. The resulting pellet was washed three times with TBS buffer, and the supernatant from these washes was designated the W fraction. The bound protein was eluted with 7% SDS (E fraction). All samples were mixed with SDS sample loading buffer

and heated at 100 °C for 2 min. Finally, the binding ability of rHev b5 to microorganisms was analyzed by 12% SDS-PAGE and western blotting with an anti-rHevb5 antibody. To confirm the contribution of divalent cations to the antibacterial activity of rHev b5, TBS without 50 mM $CaCl_2$ or $ZnCl_2$ was also tested.

## Statistical analysis

The chitinase activity and colony counting assays were conducted in triplicate, while liquid growth inhibition was performed in duplicate. Results were reported as the mean ± standard deviation. Statistical analysis of the colony counting assay was carried out using one-way ANOVA with IBM SPSS Statistics version 29.0.2.0. Duncan's multiple range tests were used to assess significant differences between mean values, with statistical significance set at $p < 0.05$.

# RESULTS

## Sequence characterization of Hev b5

The *orf* sequence, including a stop codon, of the acidic protein Hev b5 from *H. brasiliensis* under accession numbers NM_001405317.1, DQ306732.1, U51631.1, and U42640.1, showed 100% identity over 456 bp. This sequence encoded a peptide of 151 amino acids. SignalP 4.1 prediction indicated the absence of a signal peptide. The protein had a predicted isoelectric point (pI) of 4.06 and a molecular mass of 16.09 kDa.

SMART analysis revealed that the protein comprises two low complexity regions: the first region starts at position 17 and ends at position 82, and the second region starts at position 90 and ends at position 151. These regions may form surfaces for interaction with phospholipid bilayers. Both low complexity regions are rich in glutamic acid (E). Additionally, SCOPe analysis identified the d2ae2a_ domain, which starts at position 61 and ends at position 102, as being involved in classes comprising alpha and beta proteins.

The tertiary structure of Hev b5 was constructed using the SWISS-MODEL algorithm and viewed with the PyMOL program (Fig. 1). The protein was aligned with the major latex allergen Hev b5 under accession number Q39967.1, which was built using the AlphaFold v2 method. The model quality was assessed using the GMQE score, which was 0.57. The tertiary structure comprised long random coils separated by a very short strand of α-helix.

## Expression, extraction, and purification of rHev b5

The recombinant Hev b5 *orf* protein, fused with a 6xHis tag at the N-terminus, was successfully produced in *E. coli* BL21 Star (DE3) following IPTG induction, as evidenced by the prominent blue band at approximately 40 kDa in lane 2 of Fig. 2. After protein extraction, the rHev b5 bands were shown to be expressed in both the soluble and insoluble fractions (Fig. 2, lanes 3 and 4). However, only the soluble fraction was then purified using a Ni-NTA column and dialyzed to remove imidazole, resulting in purified rHev b5 with a molecular mass of 36 kDa (Fig. 2A, lane 5). Thus, the putative rHev b5 with His-tag was calculated to be 19.91 kDa, which does not match the estimated molecular mass based on its deduced amino acid sequence.

The anti-Hev b5 antibody, produced using a specific peptide of Hev b5, confirmed that the expressed bands in lanes 2 and 3, as well as the purified rHev b5 in lane 5, correspond to

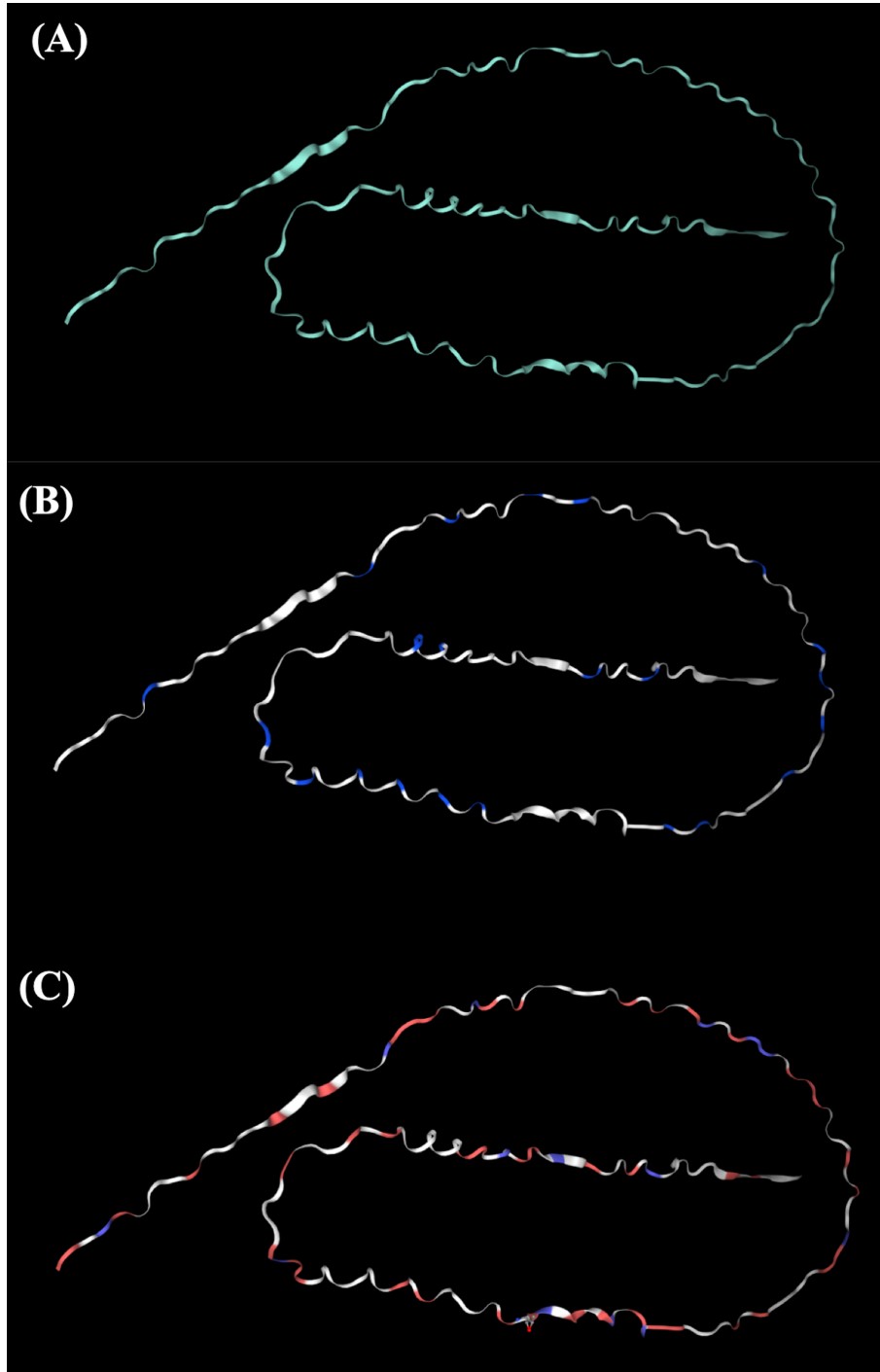

**Figure 1** **The potential tertiary structure of Hev b5 was predicted by the SWISS-MODEL program.** (A) The overall structure of rHev b5. (B) Proline, a negatively charged amino acid in protein structure of rHev b5, is indicated with blue coding, whereas white coding represents any amino acid except proline. (C) Glutamic acid and aspartic acid, which are negatively charged amino acids in the protein structure of rHev b5, are indicated with red coding, whereas blue coding was histidine, lysine and, arginine, which are positively charged amino acids and white coding represents any amino acid except positively and negatively charged amino acids.

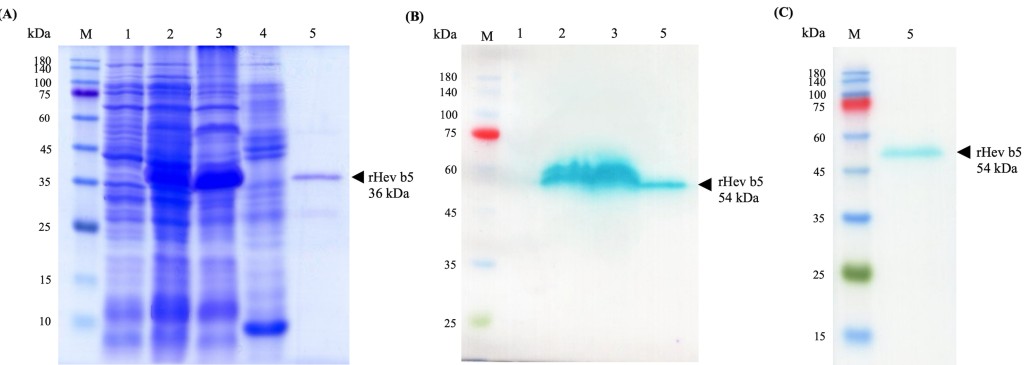

**Figure 2** **SDS–PAGE (A) and Western blot (B, C) analysis of rHev b5 from transformed BL21 Star TM (DE3)** *E. coli* **cells.** After induction with 0.5 mM IPTG at 37 °C with agitation at 200 rpm for overnight, the proteins were extracted, purified by HisTrap column and analyzed with 12% SDS-PAGE and Western blot. The arrow indicates bands of purified rHev b5. Standard protein markers indicate molecular weight in kDa (lane M). lane 1: total proteins from *E. coli* cells without IPTG induction, lane 2: total proteins from *E. coli* after induction with IPTG for overnight, lane 3: soluble protein fraction (supernatant after protein extraction with lysis buffer); lane 4: insoluble protein fraction (supernatant after resuspending the cell pellet in solubilized buffer containing 8M urea; lane 5: purified rHev b5.

rHev b5 (Fig. 2B, lane 5). However, a discrepancy was noted in the molecular weight of the band: the SDS-PAGE gel showed a band at 36 kDa, while the western blot result displayed a band at 54 kDa. This difference was further validated using an anti-His antibody (Fig. 2C, lane 5). Both the anti-Hev b5 and anti-His antibodies detected a single protein band of the same size, confirming the identity and integrity of the rHev b5 protein.

## Chitinase activity assay

The qualitative assay of chitinase activity demonstrated that rHev b5 exhibited low chitinolytic activity when screened on solid medium supplemented with 0.45% colloidal chitin as the sole carbon source, resulting in a purple color (Fig. 3A). Weak activity was achieved by 450 μg of the protein which shown as light purple. This observation was made in comparison with a positive control, which also displayed a very dark purple color, affirming the presence of moderate chitinolytic activity. In contrast, the negative control exhibited a yellow color, signifying the absence of chitinolytic activity. Additionally, the quantitative chitinase activity was evaluated using a chitinase assay kit. The results showed that the purified rHev b5 exhibited high chitinase activity at 1,723 U/mg, whereas the cell lysate of rHev b5 displayed only 36.73 U/mg. These findings suggest that rHev b5 is capable of effective enzymatic degradation of chitin, highlighting its chitinase activity in the experimental setup.

## Antifungal activity of rHev b5

The antifungal activity of rHev b5 was assessed by monitoring the growth of *P. palmivora* in PDB. After 5 days, fungal growth was measured by OD600. The results indicated that at 70 μM of rHev b5 (Fig. 3B), the OD was the lowest compared to the control group. However, no significant difference in growth was observed between the 70 and 35 μM of rHev b5

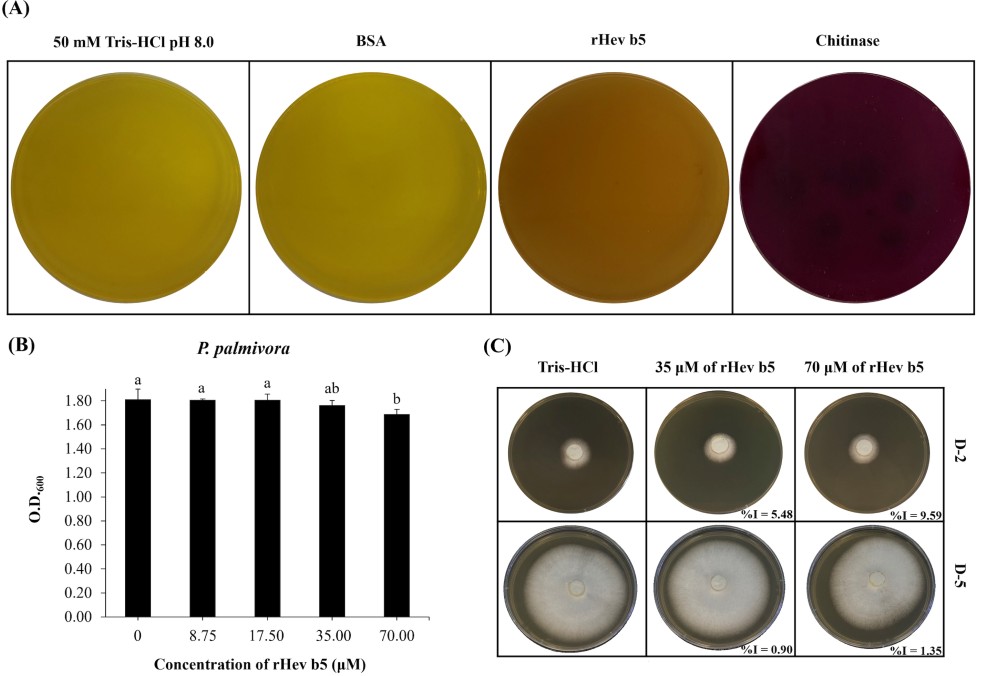

**Figure 3 Chitinolytic and antifungal activity of rHev b5 against *P. palmivora*.** (A) Qualitative assessment of rHev b5 chitinase activity on chitinase-selective medium supplemented with colloidal chitin, incubated at 25 °C for 72 h under different treatments (50 mM Tris-HCl pH 8.0, BSA, rHev b5, and chitinase). (B) Antifungal activity of rHev b5 was evaluated using a liquid inhibition assay in PDB, where *P. palmivora* was treated with rHev b5 at concentrations of 0, 8.75, 17.50, 35.00, and 70.00 μM, and optical density at 600 nm was measured after 120 h. (C) Inhibition zones of *P. palmivora* treated with 0, 35, and 70 μM rHev b5 on 1.5% (w/v) PDA, incubated at 25 °C for 2 and 5 days.

treatments. Additionally, at lower concentrations, *P. palmivora* growth was comparable to the control, suggesting that these concentrations of rHev b5 did not inhibit fungal growth after 5 days.

Since *P. palmivora* growing in PDB does not exhibit a homogeneous distribution, measuring OD600 may not clearly reflect differences in growth. Therefore, antifungal activity was further evaluated by observing radial mold growth inhibition. Figure 3C illustrates the radial growth of *P. palmivora* in the presence of 35 and 70 μM rHev b5 over 2 and 5 days. The results demonstrated that fungal inhibition was dose-dependent. At 2 days, agar plates containing rHev b5 exhibited a low inhibition percentage, ranging from 5.48% to 9.59%. However, by day 5, the inhibition had decreased to 0. 90%–1.35%.

## Antibacterial activity using liquid growth inhibition assay

The antibacterial activity of rHev b5 was assessed by measuring bacterial growth in broth media incubated with varying concentrations of rHev b5 over different time intervals. Optical density at 600 nanometers was monitored for 16 h to generate growth curves, as depicted in Fig. 4. The presence of rHev b5 slowed down the growth of both Gram-positive bacteria (*B. cereus*, *S. aureus*) and Gram-negative bacteria (*E. coli*, *S. typhi*) compared

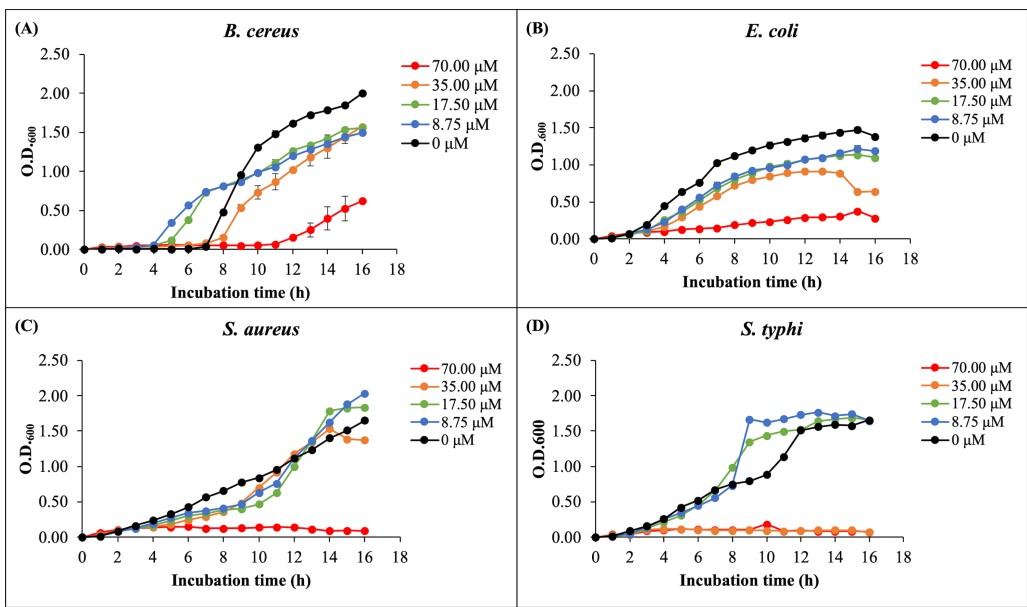

**Figure 4 The growth curves of bacteria.** (A) *B. cereus*, (B) *E. coli*, (C) *S. aureus*, and (D) *S. typhi* were monitored by measuring optical density at 600 nanometers every hour for 16 h in the presence of rHev b5 at varying concentrations (0, 8.75, 17.50, 35.00, 70.00, 0 μM).

to the control group without rHev b5. The inhibitory effect was most pronounced at a concentration of 70 μM, followed by 35, 17.5, and 8.75 μM, indicating a dose-dependent relationship between rHev b5 concentration and bacterial growth inhibition. Specifically, incubation with 70 μM rHev b5 resulted in sustained inhibition of *S. aureus* and *S. typhi* growth over the 16-hour period, similar to the effect observed with 35 μM rHev b5 on *S. typhi*. However, the inhibitory activity of rHev b5 against *B. cereus* and *E. coli* was lower. No apparent antibacterial activity was observed in the control groups.

## Antibacterial activity using colony counting method

The bacterial inhibition ability of rHev b5 was confirmed by using the colony counting method (*Tasci, 2011*). The remaining bacteria after culturing in medium containing rHev b5 were expressed as CFU/ml (as shown in Fig. 5). Comparing the groups with rHev b5 to the negative control group (without rHev b5), a significant decrease in both Gram-positive (*B. cereus*, *S. aureus*) and Gram-negative (*E. coli*, *S. typhi*) bacteria was observed. Particularly, *S. aureus* showed the most pronounced reduction with a residual bacterial count of $1.77 \times 10^3 \pm 1.15 \times 10^2$ CFU/ml, followed by *B. cereus*, *E. coli*, and *S. typhi* with residual bacterial counts of $3.50 \times 10^5 \pm 2.31 \times 10^4$, $1.01 \times 10^6 \pm 5.51 \times 10^4$, and $1.80 \times 10^6 \pm 4.16 \times 10^4$ CFU/ml, respectively. The positive control group, treated with ampicillin, exhibited the lowest residual bacterial count compared to the other groups. These results indicate that rHev b5 possesses significant bacteria inhibition ability.

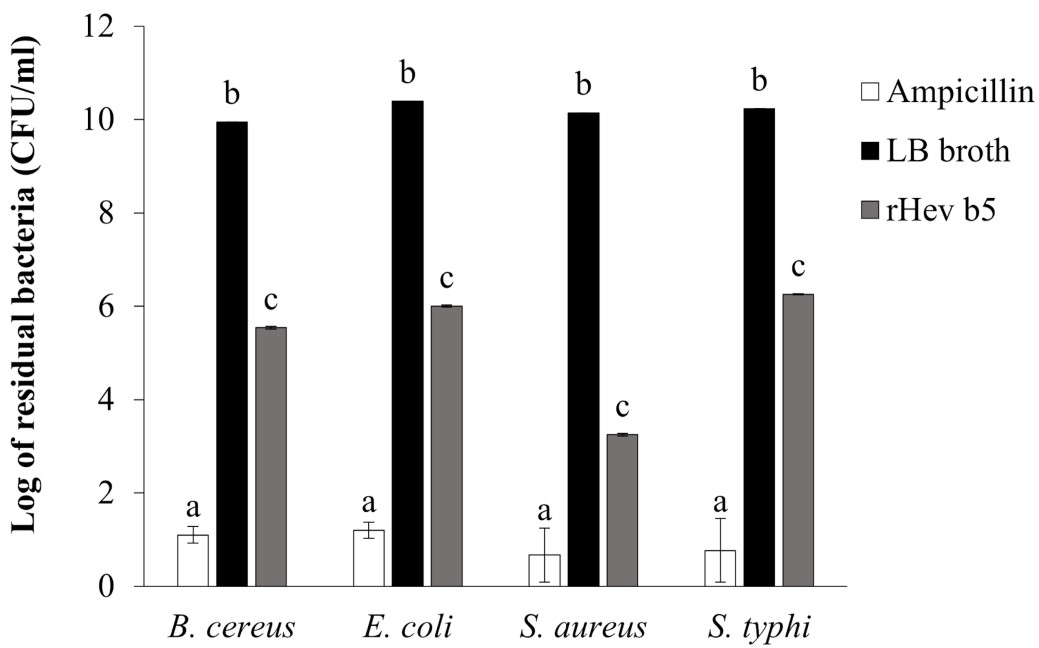

**Figure 5** **The number of bacteria in CFU/ml units.** *B. cereus*, *E. coli*, *S. aureus* and *S. typhi* bacteria were incubated with rHev b5 in LB broth at 37 °C for 12 h, with agitation at 200 rpm. Subsequently, the bacterial count was determined using the colony counting method on 1.5% (w/v) LB agar, followed by additional incubation at 37 °C for 16 h.

## Bacterial binding activity of rHev b5

Since rHev b5 possesses antimicrobial properties, its interaction with microorganisms was analyzed by assessing its binding to *S. aureus* (representing Gram-positive bacteria) and *S. typhi* (representing Gram-negative bacteria). The results showed that rHev b5 had a binding affinity for both types of bacteria. In the absence of divalent cations, rHev b5 exhibited minimal binding to the microbes. However, in the presence of calcium, its binding to both bacteria was enhanced compared to the ion-free condition (Figs. 6 and 7). Notably, the strongest binding was observed in the presence of zinc. Furthermore, rHev b5 demonstrated a significantly higher binding affinity to Gram-negative bacteria than to Gram-positive bacteria when zinc was present (Figs. 6 and 7).

## DISCUSSION

Among the various allergenic proteins, Hev b5, an acidic protein, was identified in extracts from latex gloves and shown to be an allergen (*Turjanmaa et al., 1996*). This protein stimulates histamine release from human basophils passively sensitized with serum from latex-allergic individuals in a dose-dependent manner (*Alenius et al., 1993*). Hev b5 was first discovered as a novel acidic allergen isolated from latex. It has a molecular weight of 16 kDa and an isoelectric point (pI) of 3.5. This protein is characterized by a high number of glutamic acid and proline residues arranged in a repeated motif pattern of XEEX or XEEEX, where X represents any amino acid, most frequently lysine (Lys) or alanine (Ala)

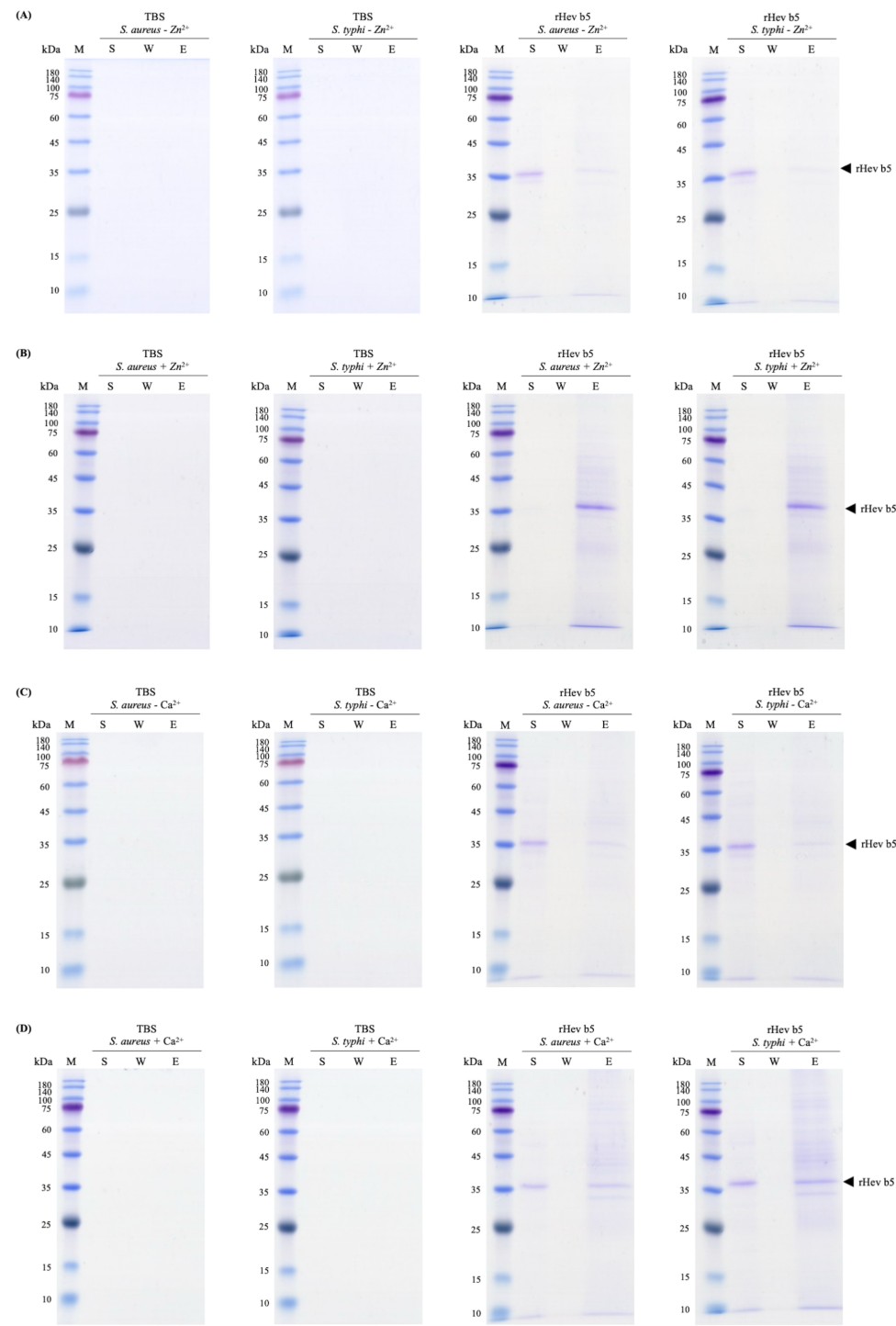

**Figure 6 SDS-PAGE analysis of each fraction from bacteria.** *S. aureus* and *S. typhi* bound to rHev b5 under conditions without or with 50 mM CaCl₂ (A, B) and 50 mM ZnCl₂ (C, D). The control group or TBS was also analyzed. The arrow indicates bands of rHev b5. Standard protein markers indicate molecular weight in kDa (lane M). Lane S: Supernatant from the centrifugation of bacteria incubated with rHev b5 at 25 °C for 20 min. Lane W: Supernatant of the cell pellet washed with TBS. Lane E: Supernatant of the cell pellet eluted with 7% SDS.

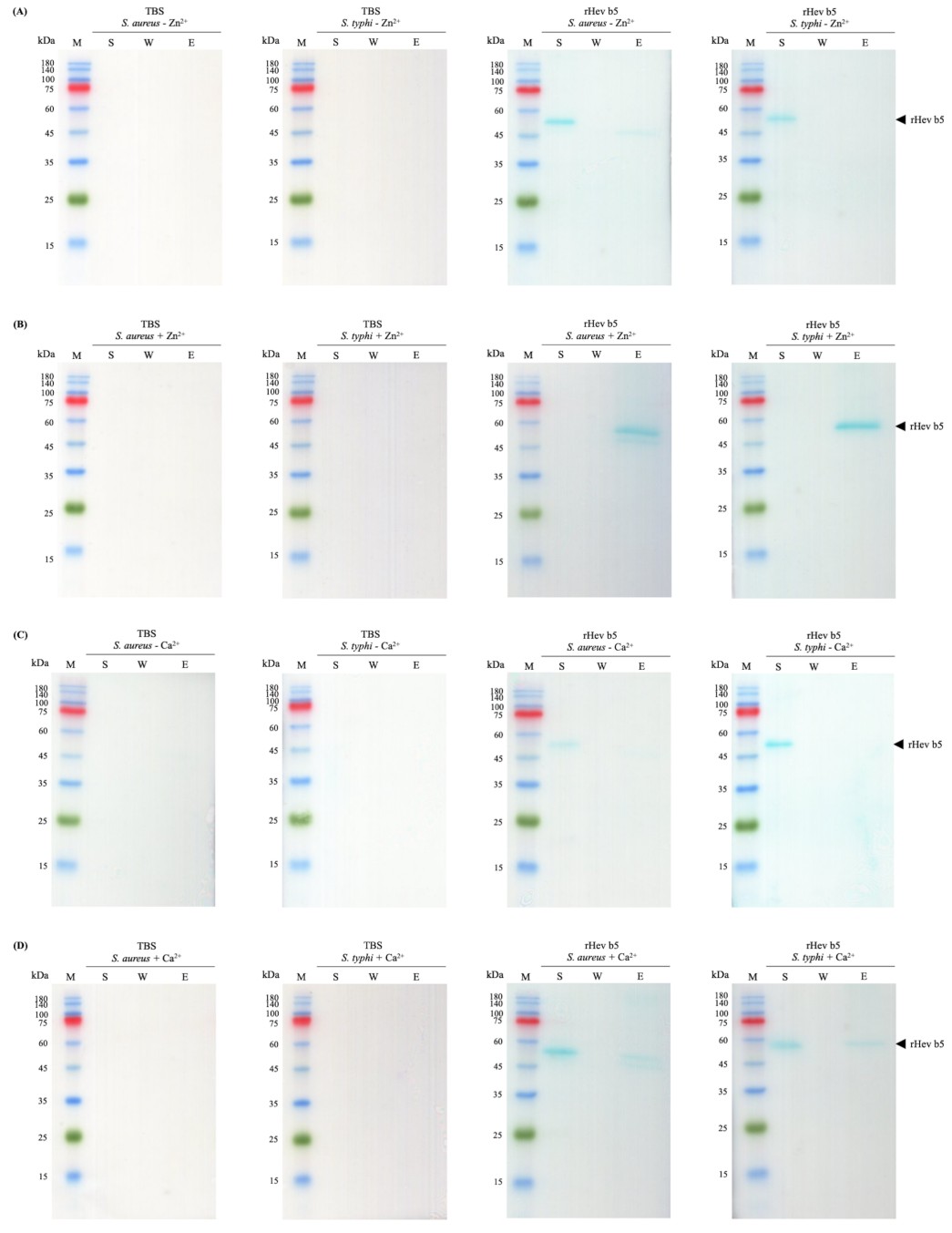

**Figure 7 Western blot analysis of each fraction from bacteria.** *S. aureus* and *S. typhi* bound to rHev b5 under conditions without or with 50 mM CaCl₂ (A, B) and 50 mM ZnCl₂ (C, D). The control group or TBS was also analyzed. The arrow indicates bands of rHev b5. Standard protein markers indicate molecular weight in kDa (lane M). Lane S: Supernatant from the centrifugation of bacteria incubated with rHev b5 at 25 °C for 20 min. Lane W: Supernatant of the cell pellet washed with TBS. Lane E: Supernatant of the cell pellet eluted with 7% SDS.

residues (*Akasawa et al., 1996*; *Slater et al., 1996*). However, its biological function remains unknown.

In this study, we described and characterized the recombinant Hev b5 allergen at both the molecular and protein levels. Hev b5, or the acidic latex protein, shares 51.22%–52.44% sequence identity with Hev b5-like proteins of *Pistacia species* (*Pistacia atlantica*, *Pistacia integerrima*, and *Pistacia vera*). The amino acid composition of Hev b5 is unusual, with a high glutamic acid content (30.5%), similar to Man e 5 from manioc (*Manihot esculenta* Crantz), which has 32% glutamic acid. Additionally, Hev b5 contains a total of 48% negatively charged residues (aspartic acid and glutamic acid) and 13% positively charged residues (arginine and lysine). The protein lacks amino acid residues, rendering it undetectable by UV spectrophotometry, a characteristic also found in glutamic acid-rich proteins in manioc.

Furthermore, SMART analysis revealed two low complexity regions that may form surfaces for interaction with the phospholipid bilayers of cells, which exhibit negatively charged phosphate groups (*Robison et al., 2016*). The tertiary structure of Hev b5 showed that most of the amino acids on the protein's surface are rich in glutamic acid, resulting in a predominantly negatively charged surface region. It is possible that the Hev b5 protein may bind with phospholipids through coordination with metal ions (*Zhu & Karlin, 1996*; *Dennison et al., 2006*).

The function of Hev b5 in *H. brasiliensis* remains unclear. To investigate its biological role, recombinant Hev b5 (rHev b5) was expressed and purified from *E. coli* with His-tag. Electrophoretic analysis revealed a band for rHev b5 at 36 kDa, which contrasts with the deduced amino acid sequence mass of 19.91 kDa. This discrepancy has also been observed in another study of rHev b5 (*Slater et al., 1996*) and in other proteins with high proline content and low isoelectric points (*Hames & Rickwood, 1990*). This is consistent with the findings of this study, where rHev b5, consisting of 187 amino acid residues-47 (25.1%) glutamic acid, 29 (15.5%) alanine, 24 (12.8%) proline, 21 (11.2%) threonine, and 10 (5.3%) serine-was observed. Notably, rHev b5 contains 11.2% proline, similar to the composition of rHev b5 reported by *Slater et al. (1996)*. Moreover, the high glutamic acid content (25%) contributes to its low pI value, leading to incomplete denaturation when analyzed using SDS-PAGE. This aligns with the study by *Slater et al. (1996)*, where rHev b5, when separated on SDS-PAGE containing 4M urea for improved separation, exhibited a molecular weight reduction from 36 to 24 kDa. Additionally, *Rasmussen (1993)* reported that proteins with collagenous domains often exhibit abnormal behavior during separation by gel filtration and SDS-PAGE, resulting in an apparent molecular weight higher than that determined by mass spectrometry and amino acid composition/sequence analyses. It has also been observed that intrinsically disordered proteins (IDPs), which lack a stable 3D structure and exhibit random coil-like behavior, bind significantly less to sodium dodecyl sulfate (SDS) compared to globular proteins. This results in abnormal migration patterns in SDS-PAGE and an overestimation of molecular weight by 1.2–1.8 times when compared to measurements obtained using mass spectrometry (*Receveur-Bréchot et al., 2006*). This phenomenon is similar to the findings in this study, where rHev b5, which possesses a random coil structure, displayed an abnormal molecular weight approximately

twice its actual size. For these reasons, it is likely that the random coil structure of rHev b5 contributes to the apparent overestimation of its molecular weight when analyzed by SDS-PAGE.

The recombinant protein was confirmed to be rHev b5 using western blotting, which relies on the specificity of antigen-antibody interactions. In this study, an anti-Hev b5 antibody was produced by Genscript using peptides specific to Hev b5. The results demonstrated that the anti-Hev b5 antibody specifically bound to the recombinant protein synthesized after IPTG induction, as observed in both cell lysate and purified rHev b5. However, the protein band observed in the western blot showed a molecular weight shift to 54 kDa (Fig. 2), likely due to aggregation during sample handling or preparation, resulting in abnormal migration and an apparent higher molecular weight on the membrane. This phenomenon has been reported by *Rath et al. (2009)*, who found that helical membrane proteins, particularly those with hairpin sequences, exhibit abnormal migration on SDS-PAGE, deviating by −10% to +30% from their actual molecular weight. The SDS-to-protein binding ratio ranged from 3.4 to 10 g of SDS per gram of protein, causing significant gel shifts. Furthermore, biophysical studies have indicated that proteins with a random coil structure are more likely to undergo aggregation due to their structural instability (*Von Bergen et al., 2005*). This observation is consistent with the findings of this study, where rHev b5, with its random coil structure (Fig. 1). Additionally, the results were further confirmed using an anti-His antibody to track rHev b5. It was found that rHev b5 could be detected at the same position (Fig. 2). Therefore, it is confirmed that the observed protein band corresponds to rHev b5, which exhibited an aberrant molecular weight during SDS-PAGE and protein transfer to the membrane.

Martin previously found that the acidic protein in latex is similar to acidic chitinase from *C. sativus* and chitinase/lysozymes from *P. quinquefolia* (*Martin, 1991*). In this study, both qualitative and quantitative analyses of chitinase activity were performed, revealing that rHev b5 exhibits clear chitinase activity. The plate containing rHev b5 demonstrated chitinase activity, which aligns with the findings of *Agrawal & Kotasthane (2012)*. They reported that colloidal chitin media containing bromocresol purple shifted towards alkalinity, causing the pH indicator dye to change from yellow to purple around the proteins. Additionally, the quantitative analysis of chitinase activity indicated that purified rHev b5 exhibited high chitinase activity, demonstrating that the chitinase activity observed in the cell lysate originated from rHev b5. Chitinase activity is crucial for breaking down chitin, a carbohydrate that forms part of the cell walls of many phytopathogens, such as fungi (*Zarei et al., 2011*), suggesting that it might act as an antifungal peptide. The antifungal activity of rHev b5 was initially evaluated by monitoring the growth of *P. palmivora* in PDB, which showed no significant inhibition after 5 days. This may be due to the non-homogeneous distribution of *P. palmivora* in the liquid culture. Therefore, antifungal activity was further assessed by measuring radial mold growth inhibition. As shown in Fig. 3C, the inhibition of *P. palmivora* radial growth was dose-dependent, particularly at 2 days, where rHev b5 exhibited a low inhibition percentage, ranging from 5.48% to 9.59% at 35 and 70 µM. However, after 5 days, the inhibition decreased to 0.90%–1.35%. This decline suggests that rHev b5 has only a very weak inhibitory effect,

which diminishes over time, eventually failing to suppress fungal growth. However, the very weak inhibition rate of rHev b5 against *P. palmivora* growth indicated that rHev b5 might degrade the cell wall of fungi through its chitinase activity. Similarly, Chi18H8, a $Ca^{2+}$-dependent mesophilic chitobiosidase, is capable of degrading chitin and also exhibits antifungal activity against phytopathogens such as *Fusarium graminearum* and *Rhizoctonia solani* (*Berini et al., 2017*).

In this study, we assessed the ability of rHev b5 to inhibit bacterial growth. The purified rHev b5 demonstrated significant inhibitory effects against all tested bacteria, including *B. cereus*, *S. aureus*, *E. coli*, and *S. typhi*, in a dose-dependent manner. Notably, increasing the concentration of rHev b5 from 35 to 70 µM resulted in nearly complete inhibition of bacterial growth in the case of *S. aureus* and *S. typhi*. However, at 70 µM, rHev b5 could inhibit the growth of *E. coli* and *B. cereus* by only about 75% (Fig. 4). The antibacterial ability of rHev b5 was confirmed using the colony counting method. The results indicated that 70 nmol of the protein could reduce the colony number of all tested microorganisms by more than 50% (Fig. 5). Notably, *S. aureus* was suppressed by more than 75% with rHev b5. Therefore, these results confirm that rHev b5 can act as an antimicrobial peptide against both Gram-positive and Gram-negative bacteria. The results indicated that rHev b5 slowed the growth of both Gram-positive bacteria (*B. cereus*, *S. aureus*) and Gram-negative bacteria (*E. coli*, *S. typhi*). It was according with *Kanokwiroon et al. (2008)* found the protein from rubber tree latex possesses the antimicrobial effects against oral pathogens especially against *Candida* spp. including *Candida albicans*, *Candida tropicalis* and *Candida krusei*.

Most structural motifs of antimicrobial peptides found in eukaryotes are alpha-helical and display cationic and amphipathic properties (*Epand & Vogel, 1999*). However, there are peptides with extended/random-coil structures that often contain a high content of proline, tryptophan, arginine, and/or histidine residues (*Takahashi et al., 2010*; *Nguyen, Haney & Vogel, 2011*). In this study, the peptide is rich in proline and glutamic acid. Therefore, Hev b5 is not only an extended/random-coil peptide but also an anionic antimicrobial peptide, possessing antimicrobial activity similar to a small portion of eukaryotic antimicrobial peptides (*Harris, Dennison & Phoenix, 2009*). Like other antimicrobial peptides, the extended peptides fold into amphipathic structures upon contact with a membrane. However, since Hev b5 contains many anionic amino acids, binding between the peptide and the phospholipid bilayer on the plasma membrane requires metal ions to form cationic salt bridges with the negatively charged components of microbial membranes (*Harris, Dennison & Phoenix, 2009*; *Dennison et al., 2006*). This study supports this mechanism, as Hev b5, an anionic antimicrobial peptide, binds with *S. aureus* and *S. typhi* in the presence of zinc or calcium ions (Figs. 6, 7 and 8). This mechanism of antimicrobial peptides induces membrane disruption, leading to cell lysis and death of microorganisms.

## CONCLUSIONS

In summary, the present study has revealed that Hev b5 exhibits chitinase activity, which provides only weak inhibition of fungal growth, while also displaying broad-spectrum antibacterial activity. It strongly binds to both Gram-positive and Gram-negative bacteria

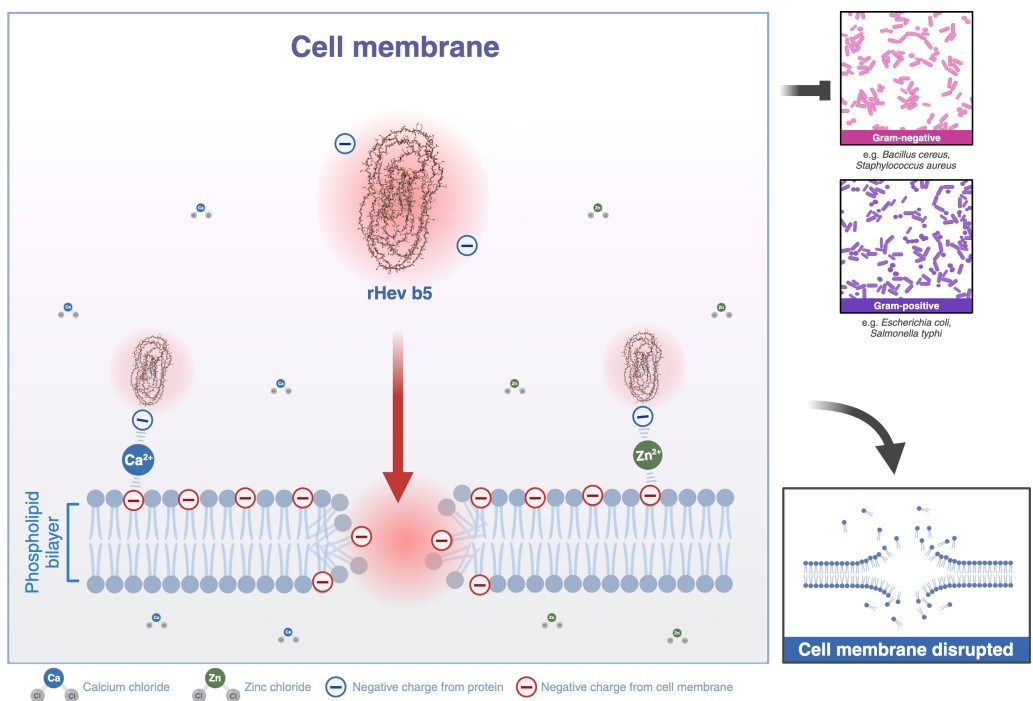

**Figure 8** **A schematic representation of the phospholipid bilayer of a cell membrane disrupted by Hev b5 in the presence of ions.** Hev b5 is an anionic antimicrobial peptide capable of interacting with the phospholipid bilayer of both Gram-positive and Gram-negative bacteria *via* zinc or calcium ion-mediated cationic salt bridges.

in the presence of divalent cations, indicating that Hev b5 may function as an anionic antimicrobial peptide.

### Funding
This research was supported by Prince of Songkla University (Grant No. SCI6505046S to Phanthipha Runsaeng). Methaphon Meethong was supported by Graduate Fellowship (Research Assistant), Faculty of Science, Prince of Songkla University, Contract no. 1-2565-02-024. The funders had no role in study design, data collection and analysis, decision to publish, or preparation of the manuscript.

### Grant Disclosures
The following grant information was disclosed by the authors:
Prince of Songkla University: SCI6505046S.
Graduate Fellowship (Research Assistant), Faculty of Science, Prince of Songkla University: 1-2565-02-024.

### Competing Interests
The authors declare there are no competing interests.

## Author Contributions

- Methaporn Meethong conceived and designed the experiments, performed the experiments, analyzed the data, prepared figures and/or tables, authored or reviewed drafts of the article, and approved the final draft.
- Kitiya Ekchaweng performed the experiments, prepared figures and/or tables, and approved the final draft.
- Sumalee Obchoei conceived and designed the experiments, analyzed the data, authored or reviewed drafts of the article, and approved the final draft.
- Chanawee Jakkawanpitak conceived and designed the experiments, performed the experiments, analyzed the data, authored or reviewed drafts of the article, and approved the final draft.
- Phanthipha Runsaeng conceived and designed the experiments, performed the experiments, analyzed the data, prepared figures and/or tables, authored or reviewed drafts of the article, and approved the final draft.

## Data Availability

The raw data is available at Zenodo: phanthipha18. (2025). phanthipha18/Hevb5: Hevb5 (Version Hevb5). Zenodo. https://doi.org/10.5281/zenodo.14897810.

## Supplemental Information

Supplemental information for this article can be found online at http://dx.doi.org/10.7717/peerj.19242#supplemental-information.

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
