# Peer review of "The acidic latex protein from Hevea brasiliensis serves as an anionic antimicrobial peptide"

_PeerJ, doi:10.7717/peerj.19242_

## Round 0.1 · original submission · Major Revisions

Three experts assessed your manuscript and found the content suitable for this journal. Several concerns were raised by the Reviewers, including the need for additional experimentation, which I also considered required (confirmation of the antifungal properties of the peptide).

Reviewer 1 ·

Basic reporting

Overall, the research paper is well focused and structured easily.
The raw data used for the analysis are provided.

However, I will make some corrections and suggestions that will help improve the work for publication.

Experimental design

In general, the methods mentioned in the work do not have references and some of them, as well as their results, need further description.

Line 41-42. Reference

lines 109-113: Was it checked for glycosylation sites?

Line 123-133: Method reference

Line 152-162: Method reference

Line 164-173: Method reference

Line 182: 25-250?

lines 175-185: Method reference

Line 242: Method reference

Line 255 : Method reference

Line 185: Please describe this method further, as well as the legend for Figure 6.

Line 266: Method reference. You can describe the method further; this will help support what you propose as the binding of the recombinant protein to the bacteria.

Validity of the findings

Line. 231: How could I solve that? Is that important for the antimicrobial property that the protein has?

Could you make the dimeric structure with AlphaFold?

Lines 276-280: Reference

Lines 299-300: Restructure these lines, since it is understood as if it were a protein from E. coli and not recombinant.


The qualitative assay of chitin's activity.
This experiment is not sufficient to conclude that it is a protein with antifungal properties. Therefore, in this part it is suggested to do other experiments, such as the growth of some pathogenic fungi in the presence of the recombinant protein. If not, this part must be restructured or not concluded.


Figure 2. Improve the figure caption description: for example, what IPTG concentration, temperature and rpm of the experiment.

How can we be sure that the antimicrobial activity is due to the protein or to some other protein of lower molecular weight? Since in the gel where the purification is shown, more proteins are observed with considerable concentrations.

Additional comments

Fig. 3. This test is the one that could perhaps be done using the recombinant protein and see its inhibitory effect on growth.

Reviewer 2 ·

Basic reporting

The research focused on evaluating the Hev b5 as a distinctive acidic protein which is reported as an allergen in natural latex and latex gloves. This protein is abundant in glutamic acid and proline, which are organized in repetitive sequences which were evaluated as an antimicrobial activity with the standard test protocols.

The basic reporting is clear, the research content is as per the research title. The English language used is as per the requirement of the journal.
The literature reference used are very relevant and are aligning with the research interest.
The figures tables, the raw data shared are appropriate and relevant to the research results obtained..

Experimental design

Experimental design illustrates the outcome and details have been shown and its aligning with the title. Research activity is well defined and proper investigation and test protocols were evaluated and tested with the available standards.
The methods which were used were well elaborated.

Validity of the findings

The results are original and robust and well supported by the experimental outcomes.

Additional comments

NA

Reviewer 3 ·

Basic reporting

Meethong et al. report the cloning, expression and characterization of Hev B5, a protein contained in latex with antimicrobial activities. Overall, it is an interesting study. However, there are points in the structure that must be addressed. The antecedents shown in the introduction are repeated in the first part of the discussion. It is necessary to restructure both sections so that the discussion mainly addresses the data generated in the work.

Lines 276-292 of the discussion section would be better placed in the introduction.

Figure 2 shows a methodological result for generating biological material with which the relevant experiments for the study will be carried out. Accordingly, the figure could be published as supplementary material.

It is desirable to improve Figure 2b; the protein band is very diffuse. Was the WB performed with all the lanes of the gel? Why is the full image not shown?

Minor concerns are listed:

Homologate and correct the nomenclature of the bacteria used; for example, Salmonella thyphi is referred to as S. Thyphi. Change to S. thyphi written in italics.

Change ORF to orf written in italics.

The constructed recombinant plasmid is referred to in two different ways, line 103 versus line 115.

The A and B portions of the serum are not specified in the introduction (line 75). Please clarify.

Experimental design

Expression, extraction, and purification of rHev b5. (lines 221-231). It is proposed that the band observed in the gel is a dimer of the protein. To prove this, a WB experiment with the purification product under denaturing conditions before dialysis would be desirable. Here, a single band of the predicted size should be visible.

Qualitative assay of chitinase activity. (lines 232-241). It is desirable to repeat the tests quantitatively. There are methodologies available that are simple. The above would better support the work.

The suggested model of action of chitinase proposes that calcium is necessary for antimicrobial activity. Accordingly, to support this effect, the tests in Figures 4 and 5 could be performed in the absence of calcium. If the proposal is correct, the antagonistic effect should be lost in this situation.

The experiment in Figure 6 is controversial. It would have been desirable to perform a WB with the gels shown. The above would confirm the presence of the protein. There are other ways to test the contact, such as performing immunodetection assays with the reported antibodies using confocal or fluorescence microscopy.

Validity of the findings

As suggested in the previous sections, the results could be better supported. The work presented in this version of the manuscript has weak points that need to be addressed. The suggested experiments and the addition of some controls would reinforce the work.

---

## Round 0.2 · accepted · Accept

The authors revised the manuscript following the Reviewers' comments. Consequently, this new version is suitable for publication.

Reviewer 2 ·

Basic reporting

Corrections are done

Experimental design

Corrections are done

Validity of the findings

Corrections are done

Reviewer 3 ·

Basic reporting

The corrected version meets the requirements

Experimental design

The corrected version meets the requirements

Validity of the findings

The corrected version meets the requirements

Additional comments

Once the recommendations have been taken into account, I consider that this new version of the article meets the quality standards of the journal to be published.